# Cervical Cancer in the Era of HPV: Translating Molecular Mechanisms into Preventive Public Health Action

**DOI:** 10.3390/ijms26178463

**Published:** 2025-08-30

**Authors:** Lidia Boldeanu, Mohamed-Zakaria Assani, Mihail Virgil Boldeanu, Isabela Siloși, Maria-Magdalena Manolea, Constantin-Cristian Văduva, Alexandru-Dan Assani, Anda Lorena Dijmărescu

**Affiliations:** 1Department of Microbiology, Faculty of Medicine, University of Medicine and Pharmacy of Craiova, 200349 Craiova, Romania; lidia.boldeanu@umfcv.ro; 2Department of Immunology, Faculty of Medicine, University of Medicine and Pharmacy of Craiova, 200349 Craiova, Romania; mohamed.assani@umfcv.ro; 3Doctoral School, University of Medicine and Pharmacy of Craiova, 200349 Craiova, Romania; alexandruassani@gmail.com; 4Department of Obstetrics and Gynecology, Faculty of Medicine, University of Medicine and Pharmacy of Craiova, 200349 Craiova, Romania; magdalena.manolea@umfcv.ro (M.-M.M.); cristian.vaduva@umfcv.ro (C.-C.V.); lorenadijmarescu@yahoo.com (A.L.D.)

**Keywords:** human papillomavirus, cervical cancer, HPV vaccination, screening and prevention, global health disparities

## Abstract

Cervical cancer remains a significant public health challenge, disproportionately affecting women in low- and middle-income countries (LMICs). Persistent infection with high-risk types of human papillomavirus (HPV), particularly HPV16 and HPV18, is the central cause of cervical carcinogenesis, driven by the viral oncoproteins E6 and E7, which disrupt the host tumor suppressors p53 and retinoblastoma protein (pRb). Advances in molecular understanding have catalyzed effective primary and secondary prevention strategies. Prophylactic HPV vaccination, especially the nonavalent formulation, has demonstrated high efficacy in reducing HPV infections and cervical precancer. Concurrently, HPV deoxyribonucleic acid (DNA) testing, self-sampling, and screen-and-treat protocols are transforming screening paradigms, particularly in resource-limited settings. However, global disparities in vaccine access, screening coverage, and health infrastructure persist, impeding progress toward the World Health Organization’s (WHO) 90–70–90 elimination targets. By synthesizing recent advances in virology, prevention strategies, and implementation innovations, such as therapeutic vaccines, artificial-intelligence (AI)-driven diagnostics, and mobile health solutions, this review sheds light on their potential to narrow these equity gaps.

## 1. Introduction

Cervical cancer remains a major global public health challenge. In 2022, approximately 660,000 new cases and 350,000 deaths were reported worldwide, making it the fourth most common cancer and cancer-related cause of death among women [1,2,3]. Alarmingly, nearly 90% of new cervical cancer cases occur in low- and middle-income countries (LMICs), largely due to inequities in access to vaccination, screening, and treatment [3]. In regions with an elevated prevalence of human immunodeficiency virus (HIV), particularly sub-Saharan Africa, women living with HIV are up to six times more likely to develop cervical cancer, underscoring the interaction of biological and social determinants [3].

At the heart of cervical cancer etiology is human papillomavirus (HPV). Over 99% of cases are linked to persistent infection with high-risk HPV (hrHPV) strains, notably HPV16 and HPV18, which together contribute to approximately 70% of invasive cancers [4,5,6,7]. While HPV infection is exceedingly common, showing global prevalence rates ranging from 11–12% overall, it is typically transient, meaning that viral DNA becomes undetectable, and cytologic abnormalities resolve without clinical intervention. Clearance is achieved through a combination of innate immune activation (including interferon release, natural killer cell activity, and local cytokine production) and adaptive immunity, particularly the development of type-specific HPV antibodies and cytotoxic T lymphocytes targeting early viral proteins (E1, E2, E6, E7). Only a fraction of persistent infections progress into precancerous lesions and, over a decade or two, invasive carcinoma [2,8,9,10,11,12]. Oncoproteins E6 and E7 play pivotal roles in this transformation by inactivating tumor suppressor p53 and retinoblastoma protein; disrupting cell cycle control; and promoting malignant progression [13,14,15,16,17,18]. Figure 1 illustrates a conceptual mind map of HPV’s process in the human organism.

Understanding this pathogenesis underpins crucial prevention strategies. Primary prevention through prophylactic HPV vaccination, particularly against types 16 and 18, has been transformative [19,20,21]. Vaccination programs are now expanding to include boys, following evidence that gender-neutral immunization enhances herd immunity and addresses HPV-associated cancers in men [20,21,22,23,24].

Secondary prevention hinges on early detection via cervical screening. Traditional cytology (pap smears) and HPV deoxyribonucleic acid (DNA) testing has reduced its incidence and mortality by up to 80% in high-income settings. However, access remains limited in resource-poor contexts. Innovative low-cost technologies, such as HPV self-sampling, point-of-care HPV testing, and portable visualization devices, have shown promise in bridging these gaps. The World Health Organization (WHO) now advocates a streamlined screen-and-treat approach tailored to low-resource settings to maximize early intervention and reduce attrition [25,26,27,28,29,30].

Despite these advances, awareness remains a critical barrier. Surveys indicate that 40% of women globally are unaware that HPV causes most cervical cancers—women lacking this awareness are less likely to undergo screening. Moreover, HPV-vaccinated individuals are, paradoxically, more likely to adhere to screening recommendations, perhaps reflecting broader engagement in preventive health behaviors [31,32,33,34].

Lastly, in 2020, the WHO launched the Global Strategy for Cervical Cancer Elimination, setting the “90–70–90” targets to be achieved by 2030: 90% of girls fully vaccinated with an HPV vaccine by age 15; 70% of women screened with a high-performance test at ages 35 and 45; and 90% of women identified with cervical disease receiving the appropriate treatment [35]. Mathematical modeling in countries like China suggests that reaching these targets, especially with adult catch-up vaccination, could drive elimination by mid-century. Yet sustained effort is needed to overcome vaccine supply shortages, screening infrastructure deficits, and socio-cultural resistance [27,35,36].

## 2. Epidemiology

### 2.1. Global Incidence and Mortality

Cervical cancer remains the fourth most commonly diagnosed cancer and the fourth leading cause of cancer death in women globally. Importantly, approximately 90% of these deaths occur in LMICs, highlighting profound global disparities [3,4,37,38,39]. This surge reflects rapid population growth in high-burden LMICs, persistently low HPV vaccination and screening coverage, high HIV co-infection rates in some regions, and entrenched socioeconomic and cultural barriers that limit access to preventive care [1,3,40,41].

### 2.2. Regional and National Trends

Incidence and mortality rates show stark regional variation. Sub-Saharan Africa bears the highest burden, followed by Central America and Southeast Asia. In contrast, North America and Western Europe maintain a relatively low incidence due to established screening and vaccination programs. In the United States (US), the age-adjusted incidence is around 7.7 per 100,000 women/year, with a death rate of 2.2 per 100,000 women/year based on Surveillance, Epidemiology, and End Results Program (SEER) data. The predictions for 2025 include 13,360 new cases and 4320 deaths. Notably, the highest incidence occurs among women aged 35–44, and more than 20% of cases involve women over 65 [3,42,43,44].

Longitudinal US data (1975–2018) show annual declines in its incidence by about 1.9% per year, slowing to −0.5% annually post-2006, while mortality has decreased by ~1% yearly. However, the incidence of distant-stage squamous cell carcinoma is rising (~1.1% per year), indicating potential delays in diagnosis [10,37,45,46,47].

### 2.3. HPV Prevalence and High-Risk Types

The prevalence of hrHPV varies geographically, reflecting population immunity, sexual behavior, and co-factors. Regionally, its prevalence is highest in sub-Saharan Africa (24%), followed by Latin America and the Caribbean (16%) and Eastern Europe and Southeast Asia (14% each) [8,42,48].

The most common hrHPV genotypes detected in the general female population are HPV16 and HPV18, which together account for about 70% of invasive cervical cancer cases worldwide [46,49,50].

In a large cohort of over 111,000 women undergoing primary HPV screening, hrHPV was detected in approximately 14% of participants—consistent with the broader range of 14–18% depending on assay sensitivity and genotyping depth [51,52].

### 2.4. Age-Specific and Histology Patterns

HPV prevalence exhibits a U-shaped age curve: peaking in younger women (<25 years), declining in mid-adulthood, and occasionally rising again post-menopause—likely due to persistent infection or cohort effects. Conversely, the incidence of cervical cancer peaks between ages 35 and 44, with a median age at diagnosis near 50 [3,37,42,43,47,53].

Histologically, squamous cell carcinoma (SCC) remains the dominant subtype, comprising ~70–80% of all invasive cervical cancers. Adenocarcinomas are increasing in many high-income regions, potentially due to the screening methods being less effective at detecting glandular precancers. In the U.S., although the overall incidence of SCC has decreased, distant-stage SCC has risen [43,45,54,55].

### 2.5. Risk Factors Beyond HPV

While hrHPV is necessary for cervical cancer development, additional co-factors influence its progression [3,41,42,56,57,58,59,60]:HIV co-infection: Women living with HIV (WLWH) have an up to six-fold higher risk of cervical cancer, partly due to compromised immunity and hrHPV persistence; globally, an estimated 5% of all cervical cancer cases are attributable to HIV co-infection;Sexual behavior: Early sexual debut, a higher number of partners, and the infection status of partners increase hrHPV acquisition;Smoking: Tobacco use contributes independently to cervical neoplasia progression;Immunosuppression: Aside from HIV, medical immunosuppression (e.g., transplant, steroids) raises its risk;Socioeconomic factors: Poverty, limited access to screening/vaccines, and education deficits amplify its risk, especially in marginalized communities.

Moreover, further co-factors that modulate the progression to high-grade lesions and invasive cancer have been mentioned, such as genetic susceptibility, hormonal influences, nutritional deficiencies, co-infections, and microbiota alterations [61,62,63,64,65,66,67,68,69,70,71,72,73,74]:Human leukocyte antigen (HLA) polymorphisms: Certain human leukocyte antigen variants may yield proteins with a lower affinity for HPV antigens, impeding immune clearance and increasing persistence risk;The TP53 codon 72 polymorphism: A variant in TP53 (P72R) has been linked to heightened vulnerability; studies report that the Arginine/Arginine (Arg/Arg) genotype is associated with increased risk of HPV-associated cervical cancer;Long-term oral contraceptive (OC) use: Among women with hrHPV infections, OC use over 5 years might double to quadruple their cervical cancer risk;High parity: A meta-analysis found that women with high parity (many full-term pregnancies) have significantly elevated odds of developing cervical cancer compared to that in low-parity women;Lower levels of vitamins A, C, E, and folate have been associated with a higher risk of cervical dysplasia and cancer, highlighting the role of antioxidant defenses in modifying HPV-related progression;Chlamydia trachomatis co-infection: Meta-analyses reveal that women with concurrent HPV and Chlamydia trachomatis infection have a substantially higher risk of cervical cancer;The depletion of protective Lactobacillus species and overrepresentation of anaerobes (e.g., Gardnerella, Prevotella, Sneathia) have been associated with higher rates of hrHPV infection and progression to precancer/cancer. Metagenomic studies observed a correlation between such dysbiosis and a higher severity of cervical lesions;An elevated vaginal pH (>5), often accompanying microbial shifts, has also been linked to a 10–20% increase in HPV positivity risk in premenopausal women.

## 3. Virology and Pathogenesis

Human papillomaviruses (HPVs) are small, non-enveloped, double-stranded DNA viruses (~8000 bp) belonging to the Papillomaviridae family. Their icosahedral capsid comprises 72 L1 pentamers and accessory L2 proteins, facilitating epithelial tropism and immune evasion, particularly via capsid stability in cutaneous and mucosal environments [75,76,77,78,79,80]. To visually consolidate the interplay between viral oncogenesis and clinical intervention, Figure 2 provides an integrated schematic of HPV-driven cervical carcinogenesis and its translational implications for screening, prevention, and future therapies.

To consolidate the significant themes of this review, Table 1 provides an integrated overview of the biological, clinical, and preventive dimensions of HPV-driven cervical cancer.

### 3.1. HPV Classification: Low-Risk vs. High-Risk

HPVs infect stratified squamous epithelia and are categorized into low-risk (lrHPV) and hrHPV based on their oncogenic potential. Low-risk types (e.g., HPV6 and HPV11) typically cause benign lesions like genital warts and low-grade dysplasia. However, rare cases of squamous cell carcinoma associated with HPV6/11 have been documented, though without evidence of viral genome integration. In contrast, high-risk types (notably HPV 16, 18, 31, 33, 45, 52, and 58) account for more than 70% of invasive cervical cancers, with HPV16 and HPV18 being most prevalent [16,88,89,90,91].

hrHPV types, most notably HPV16 and HPV18, are strongly linked to cervical cancer and other anogenital and oropharyngeal malignancies. A subset of types classified as “possible high-risk” is less frequently associated with cancer but remains under investigation due to emerging evidence. In contrast, low-risk types such as HPV6 and HPV11 are primarily associated with benign conditions, including genital warts and low-grade cervical lesions. This classification underpins both vaccine development and risk-based screening strategies [92,93,94,95]. To provide a clearer understanding of the oncogenic potential of different HPV genotypes, Table 2 categorizes HPV types by their associated clinical risk and disease outcomes.

### 3.2. Infection Mechanisms and the Viral Life Cycle

HPV infections initiate in micro-abrasions, exposing the basal keratinocytes. L1 and L2 mediate virus attachment and internalization via sulfated glycans and α6-integrins, followed by endosomal trafficking and nuclear entry. The viral genome is maintained at low copy numbers in the basal cells and replicates episomally during early infection. As infected keratinocytes differentiate, late gene expression (L1/L2) leads to virion assembly and shedding. Occasionally, HPV DNA integrates into the host genome, disrupting regulatory elements like the E2 gene and promoting oncogene overexpression [91,98,99,100].

### 3.3. Oncoproteins E6 and E7: Central Drivers of Transformation

The oncogenic potential of hrHPV is primarily mediated by the early proteins E6 and E7 [99].

E6 is a ~151-amino-acid zinc-finger protein that forms a ternary complex with host E3 ubiquitin ligase E6AP, targeting p53 for proteasomal degradation. This impairs the DNA damage response and apoptosis, allowing for the survival and proliferation of mutated cells. E6 also activates human telomerase (hTERT); interrupts PDZ-domain-containing cell polarity proteins; and dysregulates the cell cycle and differentiation [10,16,91].

E7 binds retinoblastoma protein (pRb), releasing Early Region 2 binding factor (E2F) transcription factors and promoting S-phase progression. Loss of pRb leads to unchecked cell division and genomic instability. E7 also modulates DNA damage pathways to facilitate viral replication and contributes to epigenetic reprogramming [91,101,102].

E6/E7 synergism leads to the evasion of cell cycle checkpoints, apoptosis resistance, and genomic instability—hallmarks of carcinogenesis. E5, a minor oncoprotein, further supports these processes by activating growth factor receptors [101,102,103].

The integration of hrHPV DNA into the host genome is a critical turning point toward malignancy. Disruption of the E2 regulatory region increases the expression of E6/E7, reinforcing oncogenesis. Structural studies confirm that the long control region (LCR) is altered in integrated genomes, boosting viral transcription [16,88,91,98,104].

### 3.4. Beyond E6/E7: Additional Viral and Host Factors

Recent studies highlight the emerging roles of other viral genes (like E5) and host genomic and epigenetic changes in tumor progression. Variants of HPV16 also differ in their carcinogenicity, with non-European strains showing higher risk profiles. E6/E7 expression levels correlate with gene clusters associated with aggressive phenotypes and poorer prognoses [16,102,105].

HPV16 and HPV18 are etiologically linked to the development of several epithelial malignancies, most notably cervical cancer. Their oncogenic potential is primarily driven by the coordinated actions of three early viral proteins (E5, E6, and E7), which subvert key host cellular pathways to promote viral persistence, immune evasion, and uncontrolled cell proliferation. E6 targets the tumor suppressor p53 for degradation, E7 inactivates pRb to deregulate the cell cycle, and E5 enhances growth factor signaling while impairing antigen presentation. Table 3 summarizes the primary molecular targets, biological effects, and phenotypic outcomes associated with these viral proteins [99,106,107].

## 4. Natural History and Screening

### 4.1. The Course of HPV Infection: Transient Versus Persistent

Most hrHPV infections are transient and are cleared by the immune system within 12–24 months in approximately 70–90% of women. A 2024 global meta-analysis on the genotype-specific persistence of HPV revealed that around half of persistent infections resolve within two years, validating the rationale for a 24-month surveillance approach before interventions are initiated. The natural history model of HPV infection is now understood to encompass latent states with potential reactivation, challenging the idea that a negative test equals the assumption of an all-clear [2,111,112,113,114].

In contrast, a smaller subset of infections becomes persistent, especially when involving hrHPV types, immunosuppression (e.g., HIV), smoking, or co-existing sexually transmitted infections (STIs). This persistent infection is the key precursor to cervical intraepithelial neoplasia (CIN) lesions and eventual malignant transformation [115,116,117].

### 4.2. Cervical Intraepithelial Neoplasia

CIN represents histopathological steps in cervical carcinogenesis [111,118,119,120,121,122,123,124]:CIN1 (low-grade dysplasia): Often transient and usually regresses spontaneously.CIN2/3 (high-grade dysplasia): A risk of progression to invasive carcinoma—natural history data emphasize close monitoring of CIN2 in young women due to frequent regression, while CIN3 is generally treated promptly due to a higher progression risk. Management aligns with the guidelines, balancing the risks of overtreatment versus progression to cancer.

### 4.3. Post-Treatment Surveillance

Effective post-treatment surveillance is essential for women treated for CIN2+ lesions, as these individuals remain at an elevated risk of residual or recurrent disease. hrHPV DNA testing, either alone or in combination with cytology (co-testing), is recommended due to its superior sensitivity compared to that of cytology alone, supporting its integration into follow-up protocols extending over a minimum of two years. To abridge, the natural history of hrHPV typically involves transient infection, but persistent hrHPV is the critical pivot toward CIN and cancer. Accurate detection of CIN2/3 via HPV DNA, liquid-based cytology (LBC), or visual inspection with acetic acid (VIA) remains essential. The international guidelines are shifting toward primary HPV testing with co-testing or LBC triage, and self-sampling models are increasing reach. Post-treatment, hrHPV testing ensures the early detection of recurrence, optimizing the long-term outcomes [125,126,127,128,129].

## 5. Prevention Strategies

### 5.1. HPV Vaccination

Prophylactic HPV vaccination remains the most effective primary prevention method against cervical cancer. As of 2025, 148 countries (76% of WHO member states) have included HPV vaccines in their national immunization schedules [21,130,131].

Three vaccines are widely used [132,133,134]:Bivalent (HPV-16/18);Quadrivalent (HPV-6/11/16/18);Nonavalent (HPV-6/11/16/18/31/33/45/52/58).

The nonavalent vaccine offers expanded protection and has demonstrated 97.5% efficacy against hrHPV types in a single-dose schedule. Ten-year follow-up data confirm durable immunogenicity and safety for the nonavalent vaccine in adolescents and young adults [135,136,137,138].

Innovations include single-dose schedules, which maintain ≥85% efficacy compared to that of traditional three-dose regimens. While this simplifies implementation and accessibility, global vaccination coverage remains inequitable, with catch-up programs and male inclusion lagging [136].

Table 4 summarizes the principal HPV vaccines available globally, including their valency and dosing recommendations.

Table 5 summarizes national surveillance data from countries that have achieved and sustained high HPV vaccine coverage, showing changes in the prevalence of HPV16/18, high-grade cervical lesions, and the incidence of cervical cancer in the decade following program implementation. These examples illustrate the substantial population-level benefits achieved when vaccination is introduced early, delivered before sexual debut, and maintained at high uptake.

### 5.2. Breakthrough Disease After HPV Vaccination

Although prophylactic HPV vaccination markedly reduces the incidence of high-grade cervical lesions and invasive cancer, rare cases of precancer and cancer have been documented in vaccinated individuals. These events typically occur due to one of three mechanisms [157,158]:Infection with HPV types not included in the administered vaccine;Vaccination initiated after HPV exposure;Incomplete vaccination series or an older age at first dose.

There were several documented cases and implications:A 19-year-old woman developed biopsy-confirmed CIN3 two years after completing the quadrivalent HPV vaccine series; genotyping revealed a non-vaccine HPV type [159]. Similarly, a 33-year-old woman who had received a quadrivalent vaccination in 2006 was later diagnosed with CIN3 [160]. Another report described a 30-year-old woman, vaccinated after sexual debut, who was diagnosed with adenocarcinoma in situ in the cervix—highlighting that vaccination does not treat pre-existing infections [161].Larger cohort and trial data corroborate that lesions can still emerge in vaccinated populations, albeit at much lower rates. In the Costa Rica HPV Vaccine Trial (years 7–11), vaccinated women experienced a reduction in overall high-grade disease but showed a relative increase in CIN2+/CIN3+ caused by non-vaccine HPV types, consistent with type replacement or unmasking [162]. A Swedish nationwide cohort reported an incidence of invasive cervical cancer of 47 per 100,000 in vaccinated women compared with 94 per 100,000 in unvaccinated women—confirming strong but incomplete protection [151].Importantly, the timing and completeness of vaccination influence breakthrough risk. Scottish data indicate that one or two doses at ages 12–13 confer measurable protection, whereas those at an older age generally require three doses for the optimal effect; infections and lesions can still occur during the vaccination course, especially if the exposure precedes completion [148,150].

### 5.3. Screening Programs, Screening Methods, and the Impact of Vaccination

HPV vaccination has significantly altered epidemiological patterns. In vaccinated cohorts aged 20–24 years, reports indicate declines of 79% in CIN2+ and 80% in CIN3+ from 2008 to 2022. Additionally, a nationwide U.S. study showed reduced cervical cancer incidence and mortality in girls aged 15–24 after introducing the vaccine [137,153,163,164].

Screening programs (pap and HPV testing) have similarly reduced the incidence in high-income countries by up to 80%. However, in LMICs, inadequate screening infrastructure contributes substantially to a persistently high disease burden [3,27,40,165].

The conventional pap smear has been the cornerstone of cervical screening since the mid-20th century. One review highlighted that while cytology detected high-grade lesions with fair accuracy, it underperformed in cases with concurrent inflammation or a low viral load, underscoring the limitations in isolation [115,117,166,167,168]. LBC, now the standard (e.g., ThinPrep, SurePath), improves sample adequacy, reduces artifacts, and enables reflex HPV testing [167].

HPV DNA testing directly detects high-risk viral strains and offers higher sensitivity but lower specificity than cytology. In a review, it was reported that randomized trials with HPV testing reduced the detection of CIN2+ both at 6 months and 36 months compared to that with VIA and offered better long-term reassurance. In the US, the United States Preventive Services Task Force (USPSTF) now recommends primary hrHPV testing every 5 years for women aged 30–65, with cytology alone every 3 years for ages 21–29. A 2024 update in England likewise extends HPV-based screening intervals to five years for ages 25–49, shifting to clinician-collected and self-sampling methods [169,170,171,172,173,174,175].

In some settings, co-testing (HPV+cytology) every 5 years continues, offering maximal lesion detection but increased costs and follow-up burden. The detection of hrHPV is often followed by reflex cytology to identify high-risk cases requiring colposcopy [176].

VIA remains a low-cost, single-visit option for LMICs. VIA demonstrates varying sensitivity and efficacy, which may be influenced by the administering healthcare professional. The WHO guidelines recommend integrating VIA with HPV testing or deploying screen-and-treat models for low-resource settings [27,113,166,167,171,176].

Self-collected vaginal swabs for HPV DNA testing are gaining traction: randomized trials show comparable sensitivity to that of clinician-collected samples, increasing screening uptake among underscreened women. Food and Drug Administration (FDA)-approved at-home kits offer promising access improvements, though positive results require an in-clinic follow-up [177,178,179,180]. Self-sampling has emerged as a game-changer in enhancing screening reach. A global review noted that vaginal self-swabs yielded 94.6% sensitivity and 91.6–96.8% specificity for detecting HPV, with costs reduced by 32–48%, particularly benefiting underserved populations. A U.S. intervention using the “3R” behavioral model saw 75% uptake of self-testing among medically underserved women [87,135]. Community-led initiatives, such as education culturally tailored to non-Hispanic Black women, significantly boosted uptake. A scoping review emphasized involving trusted community leaders to effectively engage indigenous, newcomer, and rural populations. In India, Accredited Social Health Activist (ASHA) workers facilitated door-to-door kit delivery under a Reach, Effectiveness, Adoption, Implementation, and Maintenance (RE-AIM) framework, enhancing its feasibility and community acceptance [181,182,183].

### 5.4. Education and Behavioral Interventions

Education and behavioral strategies are crucial to improving both vaccination and screening uptake [87,181,184,185]:Social mobilization models, such as the Information–Motivation–Behavioral Skills model, have improved community awareness and empowerment for self-sampling initiatives;Studies in the U.S. have found that women educated through culturally competent interventions were more likely to engage in self-sampling and referral follow-up;Awareness barriers persist, as only around 6.5% of some communities had heard of self-sampling, though 75% felt confident in performing it after education;Digital platforms and artificial intelligence (AI)-enhanced systems are being introduced to prompt eligible individuals, support training, and facilitate follow-up referrals;Moving forward, scaling universal vaccine coverage, enabling accessible screening, and investing in culturally informed education are essential to advancing toward the WHO target of cervical cancer elimination.

## 6. Global Disparities and Challenges

### 6.1. Inequities in Vaccine and Screening Access

Cervical cancer is a stark marker of global health inequity. Nearly 94% of cervical cancer deaths in 2022 occurred in LMICs, where screening and treatment infrastructure is lacking. While HPV vaccination has been introduced in 97% of high-income nations, only 42% of low-income countries have introduced HPV vaccination into their national immunization programs. Although vaccination programs now cover 75 countries with single-dose schedules, many remain off track to meet the WHO’s target of 90% of girls being vaccinated by age 15, a key goal of the “90-70-90” strategy [186,187,188,189].

Similarly, the screening coverage is uneven. Only 70% of women globally receive a high-performance screening test by ages 35 and 45—the second pillar of WHO’s strategy. Most LMICs fall short due to limited infrastructure, funding challenges, and a lack of trained personnel [186,187].

### 6.2. Socioeconomic, Cultural, and Health System Barriers

Barriers span several dimensions [190,191,192,193,194,195,196,197]:Low awareness of HPV and its link to cervical cancer is widespread, as seen in Ghana and Venezuela, where knowledge gaps persist even among healthcare personnel;Cultural barriers and stigma related to sexual health deter vaccine uptake and participation in pelvic exams, especially in conservative communities;Health system limitations, such as shortages of skilled pathologists and a lack of screening logistics, exacerbate inequities. For instance, Ghana relies heavily on a small number of urban-based pathologists, depriving rural areas of services.

These multifaceted barriers hinder progress toward global elimination targets.

### 6.3. Innovations Addressing Disparities

Promising strategies aim to close these gaps [30,87,186,187,188,191,198,199,200] include:Self-sampling for HPV has emerged as a scalable alternative in both high- and low-resource settings;Self-sampling is cost-effective in LMICs—studies in India and China show that combined annual self-testing and nonavalent vaccination offers superior health and economic outcomes;Mailed kits in the U.S. safety net significantly boost screening in underserved communities;The WHO-endorsed screen-and-treat model replaces multiple visits with point-of-care testing and treatment in a single visit—crucial for LMICs.

### 6.4. The WHO Cervical Cancer Elimination Initiative

Established in 2018 and formalized by the 2020 World Health Assembly, the WHO Cervical Cancer Elimination Initiative established targets of 90% HPV vaccination, 70% screening, and 90% treatment by 2030, aiming to reduce its incidence to below 4 per 100,000 women. As of mid-2025, 194 countries have endorsed the WHO strategy, and 75 have adopted single-dose schedules. However, significant gaps persist: modeling predicts that without accelerated progress, LMICs, especially in sub-Saharan Africa, will miss the 2030 benchmarks, perpetuating high mortality rates [186,187,188,190,198,199,201].

### 6.5. Steps Forward

The following steps could help close inequity gaps [186,187,188,198,200,202,203,204]:Scaling single-dose vaccination, simplifying logistics and reducing cost barriers;Expanding self-sampling, integrating it into HIV and primary care services, supported by community education;Investing in the capacity of the health system, including data systems, workforce training, and supply chain logistics;Utilizing innovative delivery models, such as mobile clinics and AI-based image analysis, to reach underserved populations;Engaging communities and women leaders, empowering them to increase the acceptance and supervision of prevention programs;Securing funding, supported by global pledges;In essence, cervical cancer disproportionally impacts women in LMICs, reflecting systemic inequities in vaccine access, screening capacity, and healthcare infrastructure. Barriers that range from cultural stigma to resource shortages compound the crisis. Nonetheless, digital health tools, self-sampling, simplified vaccination strategies, and WHO’s ambitious elimination framework offer a roadmap to equity and control. The imperative now is swift, integrated action and sustained investment.

## 7. Future Directions

As we advance toward the elimination of cervical cancer, emerging technologies and strategic innovations will enhance prevention, detection, and treatment, bridging the current gaps and charting the course for a future with a minimal disease burden.

### 7.1. Therapeutic HPV Vaccines and Immunotherapies

While prophylactic HPV vaccines prevent new infections, therapeutic vaccines aim to eliminate established HPV infections and treat high-grade lesions and cancers. Recent clinical trials are highly promising [95,198,205,206,207,208]:A novel therapeutic vaccine (Vvax001) targeting HPV16 E6/E7 showed lesion regression in 17 of 18 CIN3 patients, with half experiencing complete regression and durable HPV clearance up to 19 weeks following vaccination;In-depth reviews describe multiple platforms—peptide, DNA, ribonucleic acid (RNA), and viral/inactivated vectors—currently in phase II and III trials, designed for CIN and invasive cancer treatment;New candidates like mHTV-03E2, an messenger RNA (mRNA)-based vaccine targeting HPV-16/18, have shown potent immunogenicity and strong preclinical efficacy;Industry efforts include Transgene’s TG4001 (HPV16 E6/E7), in phase II for metastatic cervical cancers.

Alongside therapeutic vaccines, cellular therapies like TCR-engineered T cells are under exploration for refractory HPV-associated malignancies. Additionally, immune checkpoint inhibitors combined with vaccine strategies could enhance antitumor responses, potentially transforming recurrent and metastatic cervical cancer therapy [209].

### 7.2. Precision Diagnostics: Biomarkers and AI

Advances in molecular diagnostics promise more precise screening and triage [198,209,210]:DNA methylation markers and HPV integration assays offer improved specificity in distinguishing remote/transient infections from lesions with malignant potential;Liquid biopsies detecting circulating HPV DNA may enable the early detection of cancer and real-time monitoring of treatment responses;AI-based image analyses (e.g., self-supervised learning on cytology slides) enhance automated triage and extend the diagnostic capacity to low-resource settings;These advances are especially beneficial in regions with a limited presence of specialists, allowing telepathology and remote diagnostics to close care gaps.

### 7.3. Expanded Vaccination Strategies

Efforts are ongoing to optimize vaccine delivery and coverage [95,136,209,211,212]:Single-dose regimens for the nonavalent vaccine have demonstrated efficacy, offering cost-effective strategies for LMICs;Gender-neutral vaccination campaigns are gaining momentum, strengthening herd immunity and reducing non-cervical HPV cancers;Boosting vaccine equity requires immune profiling, affordable generics, cold-chain adaptation, and political commitment.

### 7.4. Self-Sampling and Decentralized Screening: The Integration of Prevention and Cancer Care, Policy, and Implementation Science

A holistic approach binds the following [85,87,95,191,212,213,214,215,216,217]:The FDA-approved Teal Health at-home HPV test delivers clinical-equivalent sensitivity and the potential to close screening gaps, particularly among underserved populations;Combining therapeutic vaccinations post-surgery may reduce recurrence rates, while observational data support its efficacy;Point-of-care integrations: Merging HIV services, self-sampling, and AI tools, focusing on high-risk groups, and maximizing the screening yield;Modeling studies emphasize that scaling the 90-70-90 strategies with adult catch-ups could lead to elimination by 2061;Essential enablers include political will, funding, cold chain logistics for vaccines, community education, and accurate data systems;By reducing the reliance on clinic-based visits, these tools democratize early detection.

The future of cervical cancer control is multidisciplinary: therapeutic vaccines, precision diagnostics, equitable vaccine deployment, self-sampling, digital automation, system integration, and an underpinning of robust policy. Together, these innovations can help realize the WHO’s vision of global elimination.

## 8. Conclusions

Cervical cancer remains a preventable malignancy, yet it continues to cause significant morbidity and mortality, especially in underserved regions. The foundation of its control rests on a three-pronged strategy: primary prevention via vaccination, secondary prevention through screening, and the equitable global implementation of both.

Despite scientific advances, global inequities persist. The WHO Cervical Cancer Elimination Initiative aims to reduce the incidence of cervical cancer to below 4 per 100,000 women annually through a 90–70–90 framework by 2030. However, its success depends critically on policy adoption, investments in the cold chain and workforce, community engagement, and strong data systems.

Looking ahead, continued innovation is essential. Emerging technologies, including therapeutic vaccines, AI-supported diagnostics, liquid biopsies, and advanced biomarkers, promise to enhance its detection, treatment, and monitoring, particularly in under-resourced settings. Likewise, integrated approaches combining vaccinations with screening offer faster progress, as demonstrated by population-level trials.

In summary, the trajectory against cervical cancer is promising and grounded in robust scientific advances and validated by decreasing disease trends. To capitalize on this momentum, global efforts must prioritize:Equitable vaccine delivery, including simplified dosing strategies and broader gender coverage;Accessible screening, leveraging self-sampling and HPV-based methods embedded into health systems;Innovative implementation, supporting diagnostics, treatment, and data systems tailored to low-resource contexts;Global solidarity, ensuring political commitment and sustained financing to close the equity gap.

## Figures and Tables

**Figure 1 ijms-26-08463-f001:**
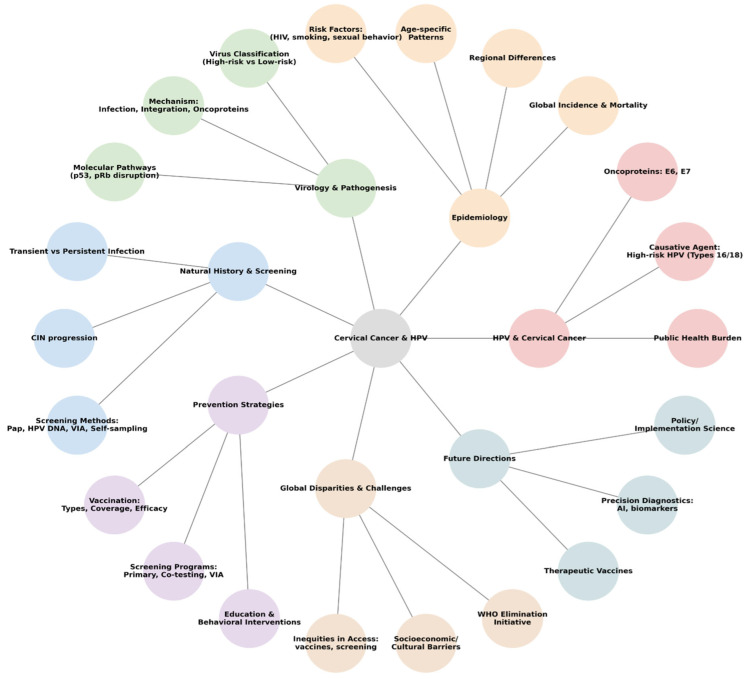
HPV: pathogenesis, prevention, and global health challenges. Conceptual mind map (figure created in BioRender). This mind map encapsulates the biological, clinical, and societal aspects of HPV. It not only covers viral pathogenesis and its link to cervical cancer but also integrates modern prevention strategies, such as vaccination and screening, while acknowledging global challenges like disparities in healthcare access and education. It is a valuable tool for health professionals, educators, and policymakers aiming to understand and communicate the comprehensive landscape of HPV control and cervical cancer elimination.

**Figure 2 ijms-26-08463-f002:**
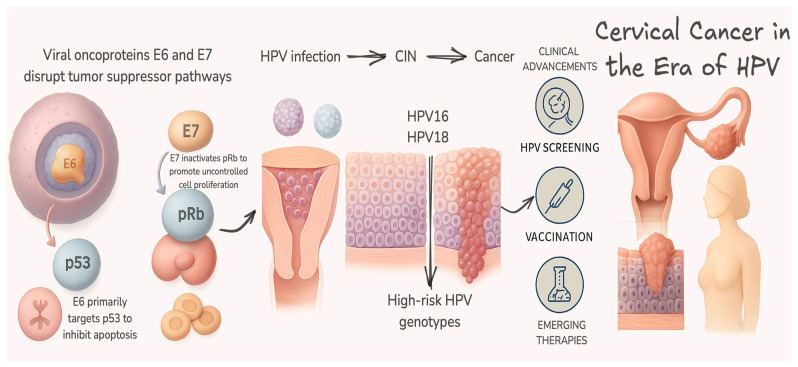
Molecular and clinical overview of human papillomavirus (HPV)-induced cervical carcinogenesis (figure created in BioRender). This schematic integrates molecular and clinical perspectives on HPV-mediated cervical cancer. Upon infection with high-risk types of HPV (16 and 18), the viral oncoproteins E6 and E7 are expressed and disrupt key tumor suppressor pathways. E6 promotes the degradation of p53, inhibiting apoptosis, while E7 inactivates retinoblastoma protein (pRb), driving uncontrolled cell proliferation. These alterations lead to the accumulation of genetic damage and progression from cervical intraepithelial neoplasia (CIN) to invasive cervical cancer. Clinically, understanding these mechanisms informs prevention and management strategies: HPV vaccination, high-performance screening, and novel therapeutic approaches targeting E6/E7 are reshaping cervical cancer control. Linking molecular pathways to public health interventions underscores the importance of translating biological insights into global elimination efforts.

**Table 1 ijms-26-08463-t001:** A summary of key molecular, clinical, and public health elements in HPV-related cervical cancer and its prevention. [1,42,81,82,83,84,85,86,87].

Domain	Component	Details/Significance
Etiology	High-Risk HPV Types	HPV16 and HPV18 account for ~70% of cervical cancer cases worldwide
	Transmission	Primarily through sexual contact; peak incidence shortly after sexual debut
Molecular Mechanism	E6 Oncoprotein	Binds E6AP to degrade p53, impairing apoptosis and DNA repair
	E7 Oncoprotein	Inactivates pRb, driving cell cycle progression and genomic instability
Natural History	Infection Progression	Mostly clears spontaneously; persistent infection may lead to CIN and invasive cancer
	Co-Factors	Immunosuppression (e.g., HIV), smoking, early sexual activity, parity, OC use
Prevention	Prophylactic Vaccines	Bivalent, quadrivalent, and nonavalent
	WHO Vaccine Target	90% of girls fully vaccinated by age 15 by 2030
Screening	Primary Testing Methods	Pap smear, HPV DNA test, co-testing, VIA
	Emerging Tools	Self-sampling, methylation biomarkers, AI-enhanced cytology
Global Disparities	LMIC Burden	90% of cervical cancer deaths occur in LMICs
	Access Gaps	Unequal access to vaccines, trained personnel, and diagnostic infrastructure
Future Directions	Therapeutic Vaccines	Target E6/E7 in persistent HPV or CIN2/3; promising in early trials
	Elimination Goals	WHO: <4 cases per 100,000 women/year (global elimination threshold)

**Table 2 ijms-26-08463-t002:** Stratification of HPV types [92,94,96,97].

Category	HPV Types
Group 1 carcinogens	16, 18, 31, 33, 35, 39, 45, 51, 52, 56, 58, 59
(carcinogenic to humans)	
Considered high-risk	
Group 2A carcinogens	68
(probably carcinogenic to humans)	
Group 2B carcinogens	26, 30, 34, 53, 66, 67, 69, 70, 73, 82, 85, 97
(possibly carcinogenic to humans)	
Group 3	6, 11
(low-risk)	

**Table 3 ijms-26-08463-t003:** Molecular targets and oncogenic mechanisms of hrHPV early proteins [99,106,107,108,109,110].

Viral Protein	Cellular Target(s)	Biological Effect	Resulting Phenotype
**E5**	EGFR, MHC-I	Enhances growth factor signaling, downregulates antigen presentation	Immune evasion, cell proliferation
**E6**	p53	Promotes ubiquitin-mediated degradation of p53	Inhibits apoptosis, impairs DNA repair
**E7**	pRb	Disrupts the pRb-E2F complex	Uncontrolled cell cycle progression

MHC-I: Major Histocompatibility Complex Class I; EGRF: epidermal growth factor receptor.

**Table 4 ijms-26-08463-t004:** Main HPV vaccines currently available worldwide [139,140,141,142,143,144,145,146,147].

Product (Maker)	HPV Types (Valency)	Label Dosing (Regulatory)
Gardasil 9 (MSD/Merck)	6, 11, 16, 18, 31, 33, 45, 52, 58 (9-valent)	Two doses (9–14 y) or three doses (≥15 y/immunocompromised) per label
Gardasil/Silgard (MSD)—quadrivalent	6, 11, 16, 18 (4-valent)	Two doses (9–13 y) or three doses (≥14 y/immunocompromised) per label
Cervarix (GSK)—bivalent	16, 18 (2-valent)	Two doses (9–14 y) or three doses (≥15 y/immunocompromised) per label
Cecolin (Innovax, China)—bivalent	16, 18 (2-valent)	Two doses (9–14 y) or three doses (≥15 y/immunocompromised) per label
Walrinva (Zerun/Walvax, China)—bivalent	16, 18 (2-valent)	Standard 2/3 doses by age
CERVAVAC (Serum Institute of India)—quadrivalent	6, 11, 16, 18 (4-valent)	Indian authorization of 2/3 doses by age

**Table 5 ijms-26-08463-t005:** Real-world impact of high-coverage HPV vaccination programs on HPV infection, cervical intraepithelial neoplasia, and cervical cancer incidence.

Country/Setting	Vaccine/Program (Cohorts)	Outcome Measured	Key Real-World Impact	Study Period
**England** **(national)** [148]	bHPV vaccine then qHPV vaccine/9vHPV vaccine; routine at 12–13 y	Cervical cancer; CIN3	Marked reductions across all deprivation groups; strongest when vaccinated at routine age	From 2006 up to 2020 follow-up
**England** **(surveillance)** [149]	Initially bHPV vaccine (Cervarix) at 12–13 y; changed to qHPV vaccine Gardasil in 2012 (HPV 16/18/6/11)	HPV16/18 infection prevalence	Around a 90% reduction in HPV16/18 in young women offered vaccination	2010–2020
**Scotland** **(national)** [150]	bHPV vaccine; high school program	CIN2 and CIN3	Reduction in pre-invasive cervical disease in vaccinated cohorts	Women aged between 20 and 60 years by 2016
**Sweden** **(national cohort)** [151]	qHPV vaccine	Invasive cervical cancer	Vaccination associated with substantially lower cervical cancer risk; strongest when vaccinated <17 y	2006–2017
**Australia** **(national, school-based)** [152]	qHPV vaccine	HPV-related disease, including high-grade abnormalities	Large population-level reductions in HPV-related outcomes; early decline in cervical abnormalities within 5 years	2007–2016
**United States** **(HPV-IMPACT, CDC)** [153]	Mixed vaccines: bHPV vaccine, qHPV vaccine and 9vHPV vaccine	CIN2 and CIN3 (screen-detected)	Among women 20–24 y, CIN2+ ↓79% and CIN3+ ↓80% from 2008 to 2022; has also declined in 25–29 y	2008–2022
**Denmark** **(population data)** [154]	National program	HPV-16/18 infection prevalence	HPV16/18 prevalence in vaccinated women fell from 15–17% pre-vaccine to <1% by 2021	2017–2024
**Norway** **(regional, real-world)** [155]	National program	CIN2+/CIN3+	CIN2+ incidence decreased markedly after initial rise; no cervical cancers recorded in vaccinated cohorts in this series	2008–2022
**Multi-country meta-analysis** [156]	Programs with high coverage	HPV infections; genital warts; CIN2+	Countries with multi-cohort, high-coverage programs show larger direct impact and herd effects; significant reductions in HPV infection and CIN2+	2014–2018

HPV—human papillomavirus; CIN2+—cervical intraepithelial neoplasia grade 2 or higher; qHPV vaccine—quadrivalent HPV vaccine (targets types 6, 11, 16, and 18); bHPV vaccine—bivalent HPV vaccine (targets types 16 and 18); 9vHPV vaccine—nonavalent HPV vaccine (targets types 6, 11, 16, 18, 31, 33, 45, 52, and 58), ↓—decrease.

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
