# Peer review of "Cervical Cancer in the Era of HPV: Translating Molecular Mechanisms into Preventive Public Health Action"

_ijms, 2025, doi:10.3390/ijms26178463_

Round 1
Reviewer 1 Report
Comments and Suggestions for Authors
The paper entitled "Cervical Cancer in the Era of HPV: Translating Molecular Mechanisms into Preventive Public Health Action", covers both basic science and public health implications of HPV-related cervical cancer.
Overall, the manuscript contains valuable and well-researched information on HPV and cervical cancer. However, the flow of the text can be challenging to follow at times. Some information appears in multiple sections, leading to redundancy, and the arrangement of topics occasionally disrupts the logical progression. A reorganization, for example, grouping related concepts together and avoiding repetition would help improve clarity, coherence, and overall readability. One possible approach could be merging Screening methods (Section 4.3) with Screening programs (Section 5.2), as they are closely related. Also, moving Impact of vaccination and screening (Section 2.6) to the Prevention strategies section (currently 5.2), so that all prevention-related topics are grouped together. (This is just one possible arrangement; other approaches could also work, but restructuring along these lines is necessary to help improving logic and readability.). Considering whether subsections 3.3.1 and 3.3.2 are necessary, as their content might be integrated into the main section without losing clarity.
Specific modifications:
- Please explain the percentages from the paragraphs (Lines 129-131; 134-135):
“The most common hrHPV genotypes found in cervical cancers are HPV-16 (≈3.2%), HPV-18 (≈1.4%), followed by HPV-52, 31, and 58. Together, HPV-16/18 are responsible for about 70% of invasive cervical cancers.”
and
“More recent genotyping reveals an overall hrHPV positivity of 15%, with rates fluctuating between 14–18% depending on assay specifics [45,46].”
Do you refer to a specific region or globally? I noticed one reference is from medRxiv, which contains non–peer‑reviewed preprints. I suggest replacing it with a peer‑reviewed source to ensure the evidence is validated.
- The section 2.5 effectively highlights the causal role of persistent high-risk HPV infection in cervical cancer, it would benefit from a more comprehensive discussion of additional cofactors that influence progression from infection to high-grade lesions and invasive disease. These include host-related factors such as genetic susceptibility (e.g., HLA variants, TP53 polymorphisms), hormonal influences (long-term oral contraceptive use, high parity), and nutritional deficiencies (low vitamins A, C, E, folate). Other important contributors are co-infections, including sexually transmitted infections such as Chlamydia trachomatis, Neisseria gonorrhoeae, Trichomonas vaginalis, HSV-2, and alterations in the vaginal microbiota (e.g., depletion of Lactobacillus and an increase in anaerobes such as Gardnerella and Atopobium, which are linked to higher risk).
Including these aspects would provide a more complete picture of cervical carcinogenesis and strengthen the manuscript’s public health relevance.
- In the section categorizing HPV types (3.1), the authors lists HPV68 as carcinogenic. However, according to the WHO Weekly Epidemiological Record (16 December 2022, No 50, pp. 645–672) — the same reference you cite — HPV68 is classified as probably carcinogenic to humans (Group 2A) rather than as definitively carcinogenic (Group 1). I recommend revising this classification or clarifying the criteria used, to ensure consistency with the cited source and WHO/IARC guidance.
- The WHO Cervical Cancer Elimination Initiative 90–70–90 is mentioned multiple times in the manuscript. I recommend avoiding repetition and improve flow.
- I suggest removing the table and the figure from the conclusion section, as this section should summarize key points shortly and find a better placement.
- To enhance clarity and help readers better visualize key concepts, I recommend incorporating more tables where appropriate. Visual summaries can improve understanding and break up dense text, especially for complex mechanisms or data.
Addressing the suggested revisions, particularly those related to clarity, organization, and referencing, will greatly enhance the quality and impact of the paper.
Author Response
We would like to sincerely thank you for your valuable comments and constructive suggestions, which have significantly contributed to improving the quality and clarity of our manuscript. We carefully considered each of your recommendations and did our best to address them thoroughly in the revised version.
Your feedback has been instrumental in helping us refine our work, and we are truly grateful for the time and effort you invested in reviewing our manuscript.
Thank you again for your thoughtful insights.
Comment 1:
Overall, the manuscript contains valuable and well-researched information on HPV and cervical cancer. However, the flow of the text can be challenging to follow at times. Some information appears in multiple sections, leading to redundancy, and the arrangement of topics occasionally disrupts the logical progression. A reorganization, for example, grouping related concepts together and avoiding repetition would help improve clarity, coherence, and overall readability. One possible approach could be merging Screening methods (Section 4.3) with Screening programs (Section 5.2), as they are closely related. Also, moving Impact of vaccination and screening (Section 2.6) to the Prevention strategies section (currently 5.2), so that all prevention-related topics are grouped together. (This is just one possible arrangement; other approaches could also work, but restructuring along these lines is necessary to help improving logic and readability.). Considering whether subsections 3.3.1 and 3.3.2 are necessary, as their content might be integrated into the main section without losing clarity.
Response 1: We have now rearranged these subheadings into 5.2
Comment 2:
Please explain the percentages from the paragraphs (Lines 129-131; 134-135):
“The most common hrHPV genotypes found in cervical cancers are HPV-16 (≈3.2%), HPV-18 (≈1.4%), followed by HPV-52, 31, and 58. Together, HPV-16/18 are responsible for about 70% of invasive cervical cancers.”
and
“More recent genotyping reveals an overall hrHPV positivity of 15%, with rates fluctuating between 14–18% depending on assay specifics [45,46].”
Do you refer to a specific region or globally? I noticed one reference is from medRxiv, which contains non–peer‑reviewed preprints. I suggest replacing it with a peer‑reviewed source to ensure the evidence is validated.
Response 2: The reference has been updated. We’ve provided a clearer explanation, but if additional clarification is needed, please let us know. (rows 144-152)
Comment 3:
The section 2.5 effectively highlights the causal role of persistent high-risk HPV infection in cervical cancer, it would benefit from a more comprehensive discussion of additional cofactors that influence progression from infection to high-grade lesions and invasive disease. These include host-related factors such as genetic susceptibility (e.g., HLA variants, TP53 polymorphisms), hormonal influences (long-term oral contraceptive use, high parity), and nutritional deficiencies (low vitamins A, C, E, folate). Other important contributors are co-infections, including sexually transmitted infections such as Chlamydia trachomatis, Neisseria gonorrhoeae, Trichomonas vaginalis, HSV-2, and alterations in the vaginal microbiota (e.g., depletion of Lactobacillus and an increase in anaerobes such as Gardnerella and Atopobium, which are linked to higher risk).
Including these aspects would provide a more complete picture of cervical carcinogenesis and strengthen the manuscript’s public health relevance.
Response 3: The suggested information has now been included. (rows 178-207)
Comment 4:
In the section categorizing HPV types (3.1), the authors lists HPV68 as carcinogenic. However, according to the WHO Weekly Epidemiological Record (16 December 2022, No 50, pp. 645–672) — the same reference you cite — HPV68 is classified as probably carcinogenic to humans (Group 2A) rather than as definitively carcinogenic (Group 1). I recommend revising this classification or clarifying the criteria used, to ensure consistency with the cited source and WHO/IARC guidance.
Response 4: We have corrected the mistake. Thank you for your observation!
Comment 5:
The WHO Cervical Cancer Elimination Initiative 90–70–90 is mentioned multiple times in the manuscript. I recommend avoiding repetition and improve flow.
Response 5: We have improved the flow by grouping the information about the initiative. If needed, we can refine the grouping further or even create a dedicated subheading for it.
Comment 6:
I suggest removing the table and the figure from the conclusion section, as this section should summarize key points shortly and find a better placement.
Response 6: We found a better placement for them. Please mention if conclusions need to be refined.
Comment 7:
To enhance clarity and help readers better visualize key concepts, I recommend incorporating more tables where appropriate. Visual summaries can improve understanding and break up dense text, especially for complex mechanisms or data.
Response 7: We have now added the following tables:
Table 1. Summary of key molecular, clinical, and public health elements in HPV-related cervical cancer and its prevention.
Table 2. HPV Types: High-Risk vs. Low-Risk
Table 3. Molecular Targets and Oncogenic Mechanisms of hrHPV Early Proteins
Table 4. Main HPV Vaccines Currently Available Worldwide
Table 5. Real-world impact of high-coverage HPV vaccination programs on HPV infection, cervical intraepithelial neoplasia, and cervical cancer incidence
Reviewer 2 Report
Comments and Suggestions for Authors
Dear Authors,
First of all, congratulations for your interesting work, the topic you present is very important. I hope that my hints will help you in the next steps of improvement and the final manuscript will be really valuable for the readers.
Minor concern, but important in the context of reading scientific paper: creating new abbreviations, unknown previously. For example, do we really need LMICs? Unlike WHO, entire name would be easier to understand. Also, some sentences are incomprehensible or probably unfinished (seems like someone was in a hurry and overlooked), double spaces, typos etc. - can you read carefully entire manuscript and correct these imperfections?
Are there any good papers presenting cases of HPV infection and precancerous or cancerous lesions after vaccinations? Or maybe during the ongoing vaccination process (more than 1 dose is required, thus, it takes time)? Although rare cases, this would be also interesting to add.
Lines 50-53 --> transcient, does it mean it disappears completely? how long does it take? should it be treated? is the immune system involved - how? This information requires further elaboration or maybe a separate paragraph.
Line 106 --> huge increase, can you explain it please? what are the reasons for this terrifying projections?
Lines 178-183 --> Consider creating a graphical depiction here.
Lines 210 - 240 --> Consider creating a graphical depiction here.
There is no explanation for nonvalent etc. Every scientific term used for the first time should be briefly explained, since not everywhere its meaning can be the same.
Which countries have obligatory HPV vaccine? How has it changed the incidence of infections, cancer, etc.? Please provide a table summarising these information (before vs after). It can be very valuable and encouraging.
Figures --> it is difficult to differentiate where the figure description ends, because it is fused with the main body of the manuscript. Separate them please. Also, some figures need further elaboration, for example Figure 1 and 2 are very schematic - as a first draft or a mindmap it is good, but way to simple for the scientific manuscript. Figure 3 definitely requires strong revision. It looks like created using AI'tools without much effort done afterwards.
Consider creating a table - comparison of all available vaccines, including their components, availability, costs, dosage etc. There are several of them in the market.
Not every abbreviation used has been explained, correct please and consider a list of abbreviations used, at the end of the paper.
Author Response
We sincerely appreciate your thorough review and insightful comments on our manuscript. Your suggestions were highly valuable and have guided us in making substantial improvements to the paper.
We have carefully addressed all the points you raised and made every effort to incorporate your feedback into the revised version. Your contribution has played an important role in enhancing the overall quality and rigor of our work.
Thank you once again for your time and thoughtful input.
Comment 1:
Minor concern, but important in the context of reading scientific paper: creating new abbreviations, unknown previously. For example, do we really need LMICs? Unlike WHO, entire name would be easier to understand. Also, some sentences are incomprehensible or probably unfinished (seems like someone was in a hurry and overlooked), double spaces, typos etc. - can you read carefully entire manuscript and correct these imperfections?
Response 1: We have checked for imperfections, but if any errors have slipped through, please let us know. An abbreviation list has been added at the end.
Comment 2:
Are there any good papers presenting cases of HPV infection and precancerous or cancerous lesions after vaccinations? Or maybe during the ongoing vaccination process (more than 1 dose is required, thus, it takes time)? Although rare cases, this would be also interesting to add.
Response 2: Added: check subheading 5.2
Comment 3:
Lines 50-53 --> transcient, does it mean it disappears completely? how long does it take? should it be treated? is the immune system involved - how? This information requires further elaboration or maybe a separate paragraph.
Response 3: Explained. If it needs any further elaboration, please mention.
Comment 4:
Line 106 --> huge increase, can you explain it please? what are the reasons for this terrifying projections?
Response 4: If any further rephrasing is needed, please let us know.
Comment 5:
Lines 178-183 --> Consider creating a graphical depiction here.
Response 5: Check Figure 2
Comment 6:
Lines 210 - 240 --> Consider creating a graphical depiction here.
Response 6: We added: Table 3. Molecular Targets and Oncogenic Mechanisms of hrHPV Early Proteins
Comment 7:
There is no explanation for nonvalent etc. Every scientific term used for the first time should be briefly explained, since not everywhere its meaning can be the same.
Response 7: Added. We referred to nonavalent, not nonvalent. Nonavalent vaccine (HPV 6/11/16/18/31/33/45/52/58)
Comment 8:
Which countries have obligatory HPV vaccine? How has it changed the incidence of infections, cancer, etc.? Please provide a table summarising these information (before vs after). It can be very valuable and encouraging.
Response 8: Please check: Table 5. Real-world impact of high-coverage HPV vaccination programs on HPV infection, cervical intraepithelial neoplasia, and cervical cancer incidence
Comment 9:
Figures --> it is difficult to differentiate where the figure description ends, because it is fused with the main body of the manuscript. Separate them please. Also, some figures need further elaboration, for example Figure 1 and 2 are very schematic - as a first draft or a mindmap it is good, but way to simple for the scientific manuscript. Figure 3 definitely requires strong revision. It looks like created using AI'tools without much effort done afterwards.
Response 9: The figures were refined. If needed we can elaborate them further.
Comment 10:
Consider creating a table - comparison of all available vaccines, including their components, availability, costs, dosage etc. There are several of them in the market.
Response 10: Created: Table 4. Main HPV Vaccines Currently Available Worldwide
Comment 11:
Not every abbreviation used has been explained, correct please and consider a list of abbreviations used, at the end of the paper.
Response 11: An abbreviation list has been added at the end.
Round 2
Reviewer 2 Report
Comments and Suggestions for Authors
I have no further comments, the Authors has addressed properly all previous comments I had. thank you.
Author Response
The revisions were made with tracking and are found in the uploaded manuscript. Thank you once again for your valuable opinion.